

# Exercise can improve sleep quality: a systematic review and meta-analysis

Masahiro Banno[1,2], Yudai Harada[2], Masashi Taniguchi[3,4], Ryo Tobita[3], Hiraku Tsujimoto[5], Yasushi Tsujimoto[6,7], Yuki Kataoka[5,8] and Akiko Noda[9,10]

[1] Department of Psychiatry, Seichiryo Hospital, Nagoya City, Aichi Prefecture, Japan
[2] Department of Psychiatry, Nagoya University Graduate School of Medicine, Nagoya City, Aichi Prefecture, Japan
[3] Division of Physical Therapy, Rehabilitation Units, Shiga University of Medical Science Hospital, Otsu City, Shiga Prefecture, Japan
[4] Department of Physical Therapy, Graduate School of Medicine, Kyoto University, Kyoto City, Kyoto Prefecture, Japan
[5] Hospital Care Research Unit, Hyogo Prefectural Amagasaki General Medical Center, Amagasaki City, Hyogo Prefecture, Japan
[6] Department of Healthcare Epidemiology, School of Public Health in the Graduate School of Medicine, Kyoto University, Kyoto City, Kyoto Prefecture, Japan
[7] Department of Nephrology and Dialysis, Kyoritsu Hospital, Kawanishi City, Hyogo Prefecture, Japan
[8] Department of Respiratory Medicine, Hyogo Prefectural Amagasaki General Medical Center, Amagasaki City, Hyogo Prefecture, Japan
[9] Chubu University Graduate School of Life and Health Sciences, Kasugai City, Aichi Prefecture, Japan
[10] Clinical Laboratory Technical Education Center, Chubu University, Kasugai City, Aichi Prefecture, Japan

Corresponding author
Masahiro Banno,
solvency@med.nagoya-u.ac.jp

## ABSTRACT

**Background**. Insomnia is common. However, no systematic reviews have examined the effect of exercise on patients with primary and secondary insomnia, defined as both sleep disruption and daytime impairment. This systematic review and meta-analysis aimed to examine the effectiveness/efficacy of exercise in patients with insomnia.

**Methods**. We searched the Cochrane Central Register of Controlled Trials, MEDLINE, Embase, PsycINFO, World Health Organization International Clinical Trials Registry Platform, and ClinicalTrials.gov to identify all randomized controlled trials that examined the effects of exercise on various sleep parameters in patients with insomnia. All participants were diagnosed with insomnia, using standard diagnostic criteria or predetermined criteria and standard measures. Data on outcome measures were subjected to meta-analyses using random-effects models. The Cochrane Risk of Bias Tool and Grading of Recommendations, Assessment, Development, and Evaluation approach were used to assess the quality of the individual studies and the body of evidence, respectively.

**Results**. We included nine studies with a total of 557 participants. According to the Pittsburgh Sleep Quality Index (mean difference [MD], 2.87 points lower in the intervention group; 95% confidence interval [CI], 3.95 points lower to 1.79 points lower; low-quality evidence) and the Insomnia Severity Index (MD, 3.22 points lower in the intervention group; 95% CI, 5.36 points lower to 1.07 points lower; very low-quality evidence), exercise was beneficial. However, exercise interventions were not associated with improved sleep efficiency (MD, 0.56% lower in the intervention group; 95% CI, 3.42% lower to 2.31% higher; moderate-quality evidence). Only four studies noted adverse effects. Most studies had a high or unclear risk of selection bias.

**Discussion**. Our findings suggest that exercise can improve sleep quality without notable adverse effects. Most trials had a high risk of selection bias. Higher quality research is needed.

# INTRODUCTION

Approximately 30% of the general population experience sleep disruption, while 10% experience both sleep disruption and daytime dysfunction consistent with a diagnosis of insomnia as defined by the National Institutes of Health (*National Institutes of Health, 2005*). Patients with insomnia are at high risk of developing hypertension, atherosclerosis, and acute myocardial infarction (*Laugsand et al., 2011*; *Fernandez-Mendoza et al., 2012*; *Nakazaki et al., 2012*). Insomnia is strongly correlated with mental illness and poses an additional risk for depression as well as suicidal ideation and behavior (*Baglioni et al., 2011*; *Bjorngaard et al., 2011*; *Pigeon, Pinquart & Conner, 2012*). Pharmacotherapy is an effective treatment for patients with insomnia. However, the use of hypnotics is associated with increased mortality (*Kripke, 2016*), and the frequency of falls and hip fractures increases when hypnotics are used in elderly individuals (*Allain et al., 2005*). Cognitive behavioral therapy for insomnia (CBT-I), the first-line treatment for insomnia (*Morin et al., 2006*), requires frequent monitoring and has a high cost (*Passos et al., 2012*).

Exercise is a nonpharmacological therapy for insomnia, is readily available, and costs less than other nonpharmacological treatments for insomnia; notably, its effects depend upon exercise type and evaluation methodology (*Youngstedt, O'Connor & Dishman, 1997*; *Driver & Taylor, 2000*; *Youngstedt, 2005*). Recent randomized controlled trials (RCTs) have confirmed that exercise has positive effects on sleep quality, sleep onset latency, total sleep time, sleep efficiency, and insomnia severity (*Passos et al., 2010*; *Reid et al., 2010*; *Hartescu, Morgan & Stevinson, 2015*). Epidemiological studies have clarified the association between exercise and decreased complaints of insomnia (*De Mello, Fernandez & Tufik, 2000*; *Youngstedt & Kline, 2006*), as well as a relationship between lower levels of physical activity and a greater prevalence of insomnia (*Morgan, 2003*). Among the main symptoms of insomnia, such as difficulty initiating sleep (DIS) or early morning awakening (EMA) (*Lichstein et al., 2003*), EMA is more frequently observed in older adults than other symptoms (*Kim et al., 2000*). These symptoms are associated with circadian core body temperature. Patients with DIS have a delayed core body temperature rhythm, whereas those with EMA have an advanced rhythm (*Lack et al., 2008*). However, experimental investigations of the effects of exercise on sleep in individuals with insomnia are lacking.

The fifth edition of the Diagnostic and Statistical Manual of Mental Disorders (DSM-5) and the third edition of the International Classification of Sleep Disorders (ICSD-3) made major revisions to their definitions of insomnia. The DSM-5 and ICSD-3 abolished the distinction between primary and secondary insomnia. The revision was based on the findings that insomnia: (1) often accompanies another disease, (2) is preceded by a

comorbid condition, (3) persists even after effective treatment for the comorbid condition, and (4) exacerbates the symptoms of the comorbid condition (*Riemann et al., 2015*). Previous systematic reviews and/or meta-analyses investigated the effects of exercise in people with sleep complaints or chronic insomnia (*Passos et al., 2012*), undefined populations (*Kubitz et al., 1996*; *Youngstedt, O'Connor & Dishman, 1997*; *Kredlow et al., 2015*), and patients with sleep problems (*Montgomery & Dennis, 2002*; *Montgomery & Dennis, 2004*; *Yang et al., 2012*). A previous review also examined the effects of exercise on sleep in specific subpopulations (e.g., cancer survivors) (*Mercier, Savard & Bernard, 2017*). However, no previous systematic reviews have examined the effect of exercise in patients with primary and secondary insomnia as defined by having both sleep disruption and daytime impairment. Investigating the effect of exercise in patients with primary and secondary insomnia would be beneficial in clinical practice since DSM-5 and ICSD-3 abolished the distinction between the two.

## Study objectives

This review aimed to examine the effects of exercise in patients with insomnia.

## MATERIALS AND METHODS

This systematic review was conducted according to the PRISMA statement (*Liberati et al., 2009*). Table S1 shows the PRISMA 2009 checklist. The detailed methods are described in CRD42016046064 in the National Institute for Health Research PROSPERO register.

### Eligibility criteria
#### Study type

We included all published and unpublished RCTs, including those that were only abstracts or letters. Crossover trials and cluster-, quasi-, and non-randomized trials were excluded. Studies in any language from any country were accepted for screening. Studies were included regardless of the follow-up period.

#### Participants

Participants included those diagnosed with insomnia using any standard diagnostic criteria such as DSM, International Classification of Diseases, ICSD, Research Diagnostic Criteria (RDC) for insomnia, or predetermined criteria and standard measures (i.e., Pittsburgh Sleep Quality Index (PSQI); *Buysse et al., 1989*), Insomnia Severity Index (ISI) (*Bastien, Vallieres & Morin, 2001*), and a sleep questionnaire). The American Academy of Sleep Medicine developed standard definitions for insomnia disorders, such as the RDC for insomnia (*Edinger et al., 2004*). We utilized the PSQI and ISI in our inclusion criteria because both are appropriate screening tools for insomnia (*Chiu et al., 2016*). All participants had insomnia-related daytime impairments or were screened using sleep questionnaires including items about such impairments. Recent RCTs were beyond the scope of this review because participants in these studies did not have insomnia-related daytime impairments (*Gebhart, Erlacher & Schredl, 2011*; *Chen et al., 2016*; *Tan et al., 2016*).

The cutoff value for the PSQI global score used to diagnose a sleep disorder was defined by the trial list. If a study did not specify a cutoff value, we surmised that a PSQI global score

>5 would be considered insomnia (*Backhaus et al., 2002*). We included patients of any age, sex, race, and setting, but excluded those with sleep apnea syndrome. We also checked the inclusion criteria for insomnia and the sleep questionnaire to determine whether the screening process selected those with daytime impairment.

### Interventions

The interventions were predetermined exercise programs. Interventions of any intensity, duration, and frequency were included. We included exercise in combination with medication if participants in the intervention and control groups were taking the same medication. We excluded interventions recommending that patients increase physical activity or encouraging improvement in self-efficacy through CBT, a mind-body bridging program, a mindfulness meditation program, massage therapy, or breathing techniques without physical activity. We examined the following interventions and comparisons:

(1)  Exercise versus non-exercise and non-medication control; and
(2)  Exercise plus medication versus medication alone.

    We excluded the following intervention: Exercise combined with another treatment (e.g., CBT).

## Outcome measures

The following primary outcomes were measured:

1.  Sleep quality according to the PSQI;
2.  Sleep efficiency defined by the percentage of time spent in bed asleep as measured objectively by a sleep device (e.g., polysomnography [PSG], actigraphy) or by reports/diaries kept by a partner or nursing staff; and
3.  Insomnia severity according to a standard measure (ISI).

    Secondary outcomes were as follows:

1.  Quality of life (QOL) as measured by standardized questionnaires with established reliability and validity, such as the Short Form 36 (SF-36);
2.  Sleep onset latency as measured objectively by sleep devices (e.g., PSG, actigraphy) or reports/diaries maintained by a partner or nursing staff;
3.  Total sleep time as measured objectively by a sleep device (e.g., PSG, actigraphy) or reports/diaries maintained by a partner or nursing staff;
4.  All adverse events (defined by the trial list);
5.  Sleepiness during daily life according to a self-report using a standardized measure, e.g., the Epworth Sleepiness Scale (ESS);
6.  Current sleepiness according to a self-report using a standardized measure, e.g., the Stanford Sleepiness Scale (SSS);
7.  Wake after sleep onset (WASO) as measured objectively by a sleep device (e.g., PSG, actigraphy) or reports/diaries maintained by a partner or nursing staff;
8.  Anxiety according to a standardized questionnaire with established reliability and validity (e.g., State-Trait Anxiety Inventory); and
9.  Depression according to a standardized questionnaire with established reliability and validity (e.g., Beck Depression Inventory).

We consulted an expert in sleep medicine (AN) and experts in exercise therapy (MT and RT) and selected the moderator (primary and secondary outcomes, prioritization of outcomes, and subgroup analysis items) in terms of clinical importance.

## Search methods for study identification
### Electronic searches
To identify relevant trials, we searched the following electronic databases on October 9, 2016 and updated the electronic searches on October 4, 2017:

1. The Cochrane Central Register of Controlled Trials (CENTRAL);
2. MEDLINE via EBSCOhost;
3. Embase; and
4. PsycINFO via PsycNET.

See Appendix S1 for details about the search strategies.

### Searches of other resources
We also searched the following registries to identify completed but unpublished trials and investigate reporting bias.

1. World Health Organization International Clinical Trials Registry Platform; and
2. ClinicalTrials.gov.

See Appendix S1 for details of the search strategies.

We also manually searched reference lists in clinical guidelines on exercise for insomnia and in related guidelines (*Morgenthaler et al., 2006*; *Bauer et al., 2007*; *Wilson et al., 2010*; *NICE, 2012*; *Bauer et al., 2013*; *NICE, 2013*; *University of Texas at Austin School of Nursing, 2014*; *Bauer et al., 2015*; *NICE, 2015*; *Qaseem et al., 2016*), reference lists of extracted studies, and articles citing the extracted studies.

We contacted authors if the extracted studies lacked the necessary information.

## Data collection and analysis
### Study selection
Two of the five authors (MB, YH, HT, YT, and YK) independently screened the titles and abstracts of the articles identified in the search. Two of the five authors were assigned to each article to reduce the burden on each author. They assessed eligibility based on a full-text review. Disagreement was resolved by discussion; if necessary, YK or YT (if YK and an author other than YT were the two authors) or MB (if YK and YT were the two authors) provided arbitration. We followed a pre-defined protocol to screen the abstracts and full texts and used pre-defined criteria in the registered protocol. One lead author (MB) checked all included studies and the exclusion criteria for all records subjected to the full-text screening procedure. Therefore, the decision would not differ systematically.

### Data extraction and management
The data were extracted on prespecified forms that were piloted using a random sample of 10 studies. Two of the four authors (MB, HT, YT, and YK) independently extracted the data. MB and another author were assigned to each article to reduce the burden on each author. We contacted the authors of studies lacking sufficient information as necessary. Differences in data extraction opinions were resolved by discussion and arbitrated by YK

or YT (if YK was the other author) when necessary. See Appendix S2 for details of the extracted information.

### Assessment of risk of bias of the included studies
Two of the four authors (MB, HT, YT, and YK) independently assessed the risk of bias of the included studies using the Cochrane Risk of Bias Tool (*Higgins & Green, 2011*). MB and another author were assigned to each article to reduce the burden on each author. Differences in opinion about the assessment of risk of bias were resolved by discussion and through arbitration by YK or YT (if YK was the other author) as necessary.

### Measures of treatment effect
For continuous outcomes (sleep quality, sleep efficiency, insomnia severity, QOL, sleep onset latency, total sleep time, sleepiness during daily life, current sleepiness, WASO, anxiety, and depression), the standardized mean difference (SMD) or mean difference (MD) with 95% CI was calculated as recommended by the Cochrane handbook (*Higgins & Green, 2011*). We used MD when data including meta-analyses were derived from the same indicator. We used SMD when data including meta-analyses were derived from different indicators or we compared the data in the meta-analysis with data in a previous study using SMD. Adverse events were narratively summarized since the definition of these outcomes varied among studies.

### Assessment of heterogeneity
We first assessed heterogeneity by visual inspection of the forest plots. We also calculated $I^2$ statistics and analyzed them according to recommendations in the Cochrane handbook (0–40%, might not be important, 30–60%, may represent moderate heterogeneity, 50–90% may represent substantial heterogeneity, and 75–100% may represent considerable heterogeneity). When heterogeneity was detected ($I^2 > 50\%$), we attempted to identify possible causes.

### Data synthesis
We pooled the data using a random-effects model. The DerSimonian and Laird method was used in the random-effects meta-analysis (*DerSimonian & Laird, 1986*). All analyses were conducted using Review Manager software (RevMan 5.3; The Nordic Cochrane Centre, The Cochrane Collaboration, Copenhagen, Denmark).

### Subgroup analysis and investigation of heterogeneity
We further aimed to identify possible causes of heterogeneity. The following prespecified subgroup analyses of the primary outcomes were planned: (1) sex; (2) primary and secondary insomnia; (3) exercise duration: short-term (<2 months), medium-term (2 to <6 months), long-term (≥6 months); (4) exercise intensity: aerobic versus anaerobic exercise; (5) exercise type: aerobic (walking), resistance, and aerobic plus resistance; and (6) exercise setting or location: home, physical therapy center, hospital, or elsewhere.

**Table 1 Summary of findings**

| Outcomes (time frame) | Number of participants (studies) in follow-up | Quality of evidence (GRADE) | Relative effect (95% CI) | Anticipated absolute effects[a] (95% CI) | |
|---|---|---|---|---|---|
| | | | | Risk with control | Risk difference with exercise |
| Total PSQI score (8 wks to 6 mos) Scale: 0 to 21 | 361 (6 RCTs) | ⊕⊕⊖⊖ LOW[b,c,d] | – | | MD 2.87 point lower (3.95 lower to 1.79 lower) |
| Sleep efficiency (%) (1 d to 6 mos) assessed with: polysomnography and actigraphy Scale: 0 to 100 | 186 (4 RCTs) | ⊕⊕⊕⊖ MODERATE[d] | – | | MD 0.56% lower (3.42 lower to 2.31 higher) |
| Total ISI score(4–6 mos) Scale: 0 to 28 | 66 (2 RCTs) | ⊕⊖⊖⊖ VERY LOW[b,c,d,e,f] | – | | MD 3.22 point lower (5.36 lower to 1.07 lower) |
| Sleep onset latency (minute) (1 d to 6 mos) | 206 (5 RCTs) | ⊕⊕⊖⊖ LOW[d,g] | – | | MD 1.9 minutes higher (3.63 lower to 7.43 higher) |
| Total sleep time (minute) (1 d to 6 mos) | 206 (5 RCTs) | ⊕⊕⊖⊖ LOW[d,g] | – | | MD 4.32 minutes higher (9.19 lower to 17.84 higher) |
| All adverse events (2–6 mos) | 150 (4 RCTs) | ⊕⊖⊖⊖ VERY LOW[b,c,d,h,i] | – | | |

**Notes.**

[a]The risk in the intervention group (and its 95% CI) is based on the assumed risk in the comparison group and the relative effect of the intervention (and its 95% CI).

[b]Participants were not blinded.

[c]The outcome assessors were not blinded.

[d]Sample size was small. Sample size did not meet criteria of optimal information size (OIS) (400). OIS was 400 if alpha = 0.05, beta = 0.2, delta = 0.2.

[e]Allocation concealment was not done in 40% of participants.

[f]There were incomplete outcome data in 40% of participants.

[g]There were incomplete outcome data in 25% of participants.

[h]There were incomplete outcome data in 50% of participants.

[i]Allocation concealment was not done in 30% of participants.

ISI, Insomnia Severity Index; MD, mean differences; OIS, optimal information size; GRADE, Grading of Recommendations, Assessment, Development, and Evaluation; OR, odds ratio; PSQI, Pittsburgh Sleep Quality Index; RCTs, randomized controlled trials; RR, risk ratio.

GRADE working group grades of evidence: High quality, We are very confident that the true effect lies close to that of the estimate of the effect; Moderate quality, We are moderately confident in the effect estimate: The true effect is likely to be close to the estimate of the effect, but a substantial difference is possible; Low quality, Our confidence in the effect estimate is limited: The true effect may be substantially different from the estimate of the effect; Very low quality, We have very little confidence in the effect estimate: The true effect is likely to be substantially different from the estimate of effect.

### Sensitivity analysis

The following prespecified sensitivity analyses of the primary outcomes were planned: (1) repeating the analysis but restricting it to studies with low risks of bias from random sequence generation and allocation concealment, using the Cochrane Risk of Bias Tool (*Higgins & Green, 2011*); (2) repeating the analysis using a fixed-effects model instead of random-effects model; and (3) excluding studies with "a per-protocol analysis" or "analysis including imputed data."

### Summary of findings tables

The main results of our review are presented in the Summary of findings table (Table 1), which includes an overall grading of the evidence related to each of the main outcomes using

the Grading of Recommendations, Assessment, Development, and Evaluation (GRADE) approach (*Guyatt et al., 2011*; *Higgins & Green, 2011*).

### Registration

We registered the protocol in the National Institute for Health Research PROSPERO register (http://www.crd.york.ac.uk/PROSPERO/display_record.asp?ID=CRD42016046064).

## RESULTS

### Search results

After removing duplicates, we identified 4,085 records during the search conducted in October 2016 and updated the electronic searches on October 4, 2017 (Fig. 1). We included 17 trials in the qualitative synthesis and detected seven unpublished trials and one completed trial without outcomes data (*Chan et al., 2017*). Ultimately, 557 participants in nine trials were included in the quantitative synthesis.

Table 2 summarizes the published studies included in the qualitative synthesis. Table S2 shows the characteristics of the seven unpublished trials. Table S3 shows the sleep medications used in the included completed trials.

The bias risk of the quantitative synthesis is shown in Figs. 2A and 2B.

### Primary outcomes
#### Sleep quality

Data from six trials comprising 361 participants that measured sleep quality were pooled in our meta-analysis (*Reid et al., 2010*; *Tang, Liou & Lin, 2010*; *Irwin et al., 2014*; *Hartescu, Morgan & Stevinson, 2015*; *Chan et al., 2016*; *Tadayon, Abedi & Farshadbakht, 2016*) (Fig. 3A). All trials measured PSQI and had an intervention period of eight weeks to six months. There was a significant effect noted in favor of the intervention (MD, 2.87 points lower in the intervention group; 95% CI, 3.95 points lower to 1.79 points lower; $P < 0.001$; low-quality evidence). A lower score was more beneficial in PSQI. Substantial heterogeneity was observed (Tau$^2$ = 1.18; $I^2$ = 68%).

#### Sleep efficiency

Data from four trials that examined sleep efficiency in 186 participants were pooled in our meta-analysis (*Passos et al., 2010*; *Afonso et al., 2012*; *Irwin et al., 2014*; *Hartescu, Morgan & Stevinson, 2015*) (Fig. 3B). All trials measured sleep efficiency with PSG and actigraphy and had an intervention period of 1 day to 6 months. There was no significant improvement in favor of the intervention (MD, 0.56% lower in the intervention group; 95% CI, 3.42% lower to 2.31% higher; $P = 0.70$; moderate-quality evidence). A higher percentage was more beneficial for sleep efficiency. No statistical heterogeneity was indicated (Tau$^2$ <0.001; $I^2$ = 0%).

#### Insomnia severity

Data from two trials that measured insomnia severity in 66 participants were pooled in our meta-analysis (*Afonso et al., 2012*; *Hartescu, Morgan & Stevinson, 2015*) (Fig. 3C). All trials measured ISI and had an intervention period of four to six months. There was significant

**Table 2  Summary of the published studies including qualitative synthesis.**

| Source | Setting | Patients, N | Age | Inclusion criteria | Exercise type | Exercise frequency | Exercise duration |
|---|---|---|---|---|---|---|---|
| *Afonso et al. (2012)* | Elsewhere | 61 | 50 to 65 years | Postmenopausal women with primary insomnia meeting DSM-4 | Aerobic (other aerobic) | 2 session/wk | 4 mos |
| *Chan et al. (2016)* | Elsewhere | 52 | 60 years or older | Older adults with cognitive impairment with CPSQI >5 | Aerobic (other aerobic) | 2 session/wk | 2 mos |
| *Chan et al. (2017)* | Elsewhere | Unknown | 18 years or older | Participants with mild to moderate depression and PSQI >5 | Aerobic (other aerobic) | 3 session/wk | 8 wks |
| *Guilleminault et al. (1995)* | At home | 32 | 34 to 55 years | Patients with psychophysiologic insomnia meeting predetermined criteria | Aerobic (walking) | 7 d/wk | 4 wks |
| *Hartescu, Morgan & Stevinson (2015)* | At home | 41 | 40 years or older | Inactive adults meeting RDC for insomnia | Aerobic (walking) | 5 d/wk | 6 mos |
| *Irwin et al. (2014)* | Elsewhere | 123 | 34 to 55 years | Older adults with chronic and primary insomnia meeting DSM-IV-TR and ICSD-2 | Aerobic (other aerobic) | 1 d/wk | 4 mos |
| *Passos et al. (2010)* | Exercise laboratory | 48 | 30 to 55 years | Primary insomnia meeting DSM-IV-TR and ICSD-2 | Aerobic (walking, other aerobic) | Acute | One time |
| *Reid et al. (2010)* | Elsewhere | 17 | 55 years or older | Older adults with insomnia meeting predetermined criteria | Aerobic (walking, other aerobic) | 4 times per wk | 16 wks |
| *Tadayon, Abedi & Farshadbakht (2016)* | At home | 112 | Mean 52.39 (SD 1.65) years | Postmenopausal women with PSQI >5 | Aerobic (walking) | 7 d/wk | 12 wks |
| *Tang, Liou & Lin (2010)* | At home | 71 | Mean 51.80 (SD 12.13) years | Cancer patients with PSQI >5 | Aerobic (walking) | 3 d/wk | 8 wks |

**Notes.**

*Chan et al. (2017)* was included in the qualitative synthesis but excluded in the quantitative synthesis because the trial did not include outcomes data for a meta-analysis.

DSM, Diagnostic and Statistical Manual of Mental Disorders; ICSD, International Classification of Sleep Disorders; PSQI, Pittsburgh Sleep Quality Index; RDC, research diagnostic criteria.

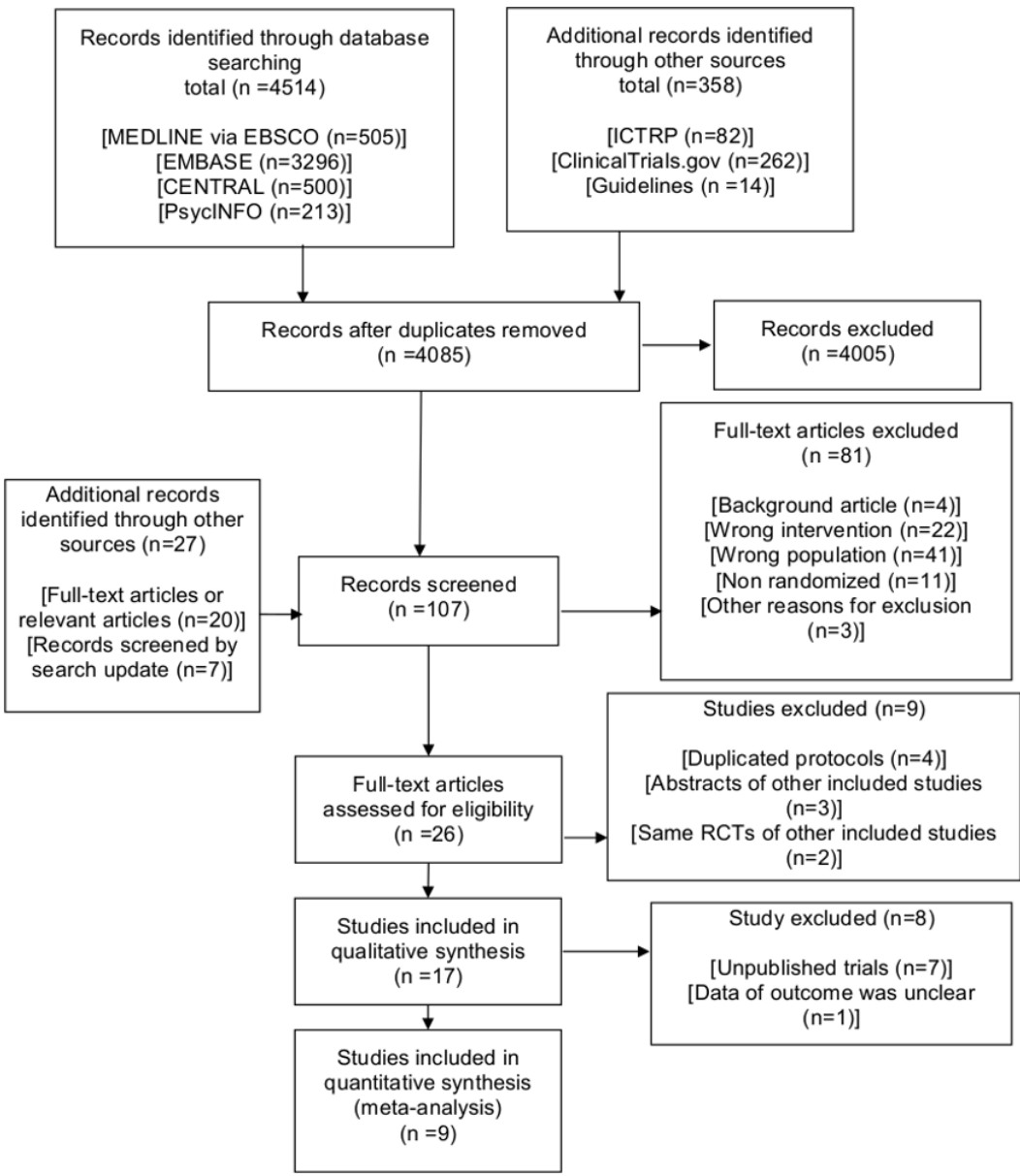

**Figure 1 PRISMA 2009 flow diagram.** CENTRAL: Cochrane Central Register of Controlled Trials; IC-TRP, International Clinical Trials Registry Platform; RCTs, randomized controlled trials

improvement in favor of the intervention (MD, 3.22 points lower in the intervention group; 95% CI, 5.36 points lower to 1.07 points lower; $P = 0.003$; very low-quality evidence). A lower score was more beneficial in ISI. No statistical heterogeneity was indicated (Tau$^2$ <0.001; $I^2 = 0$%).

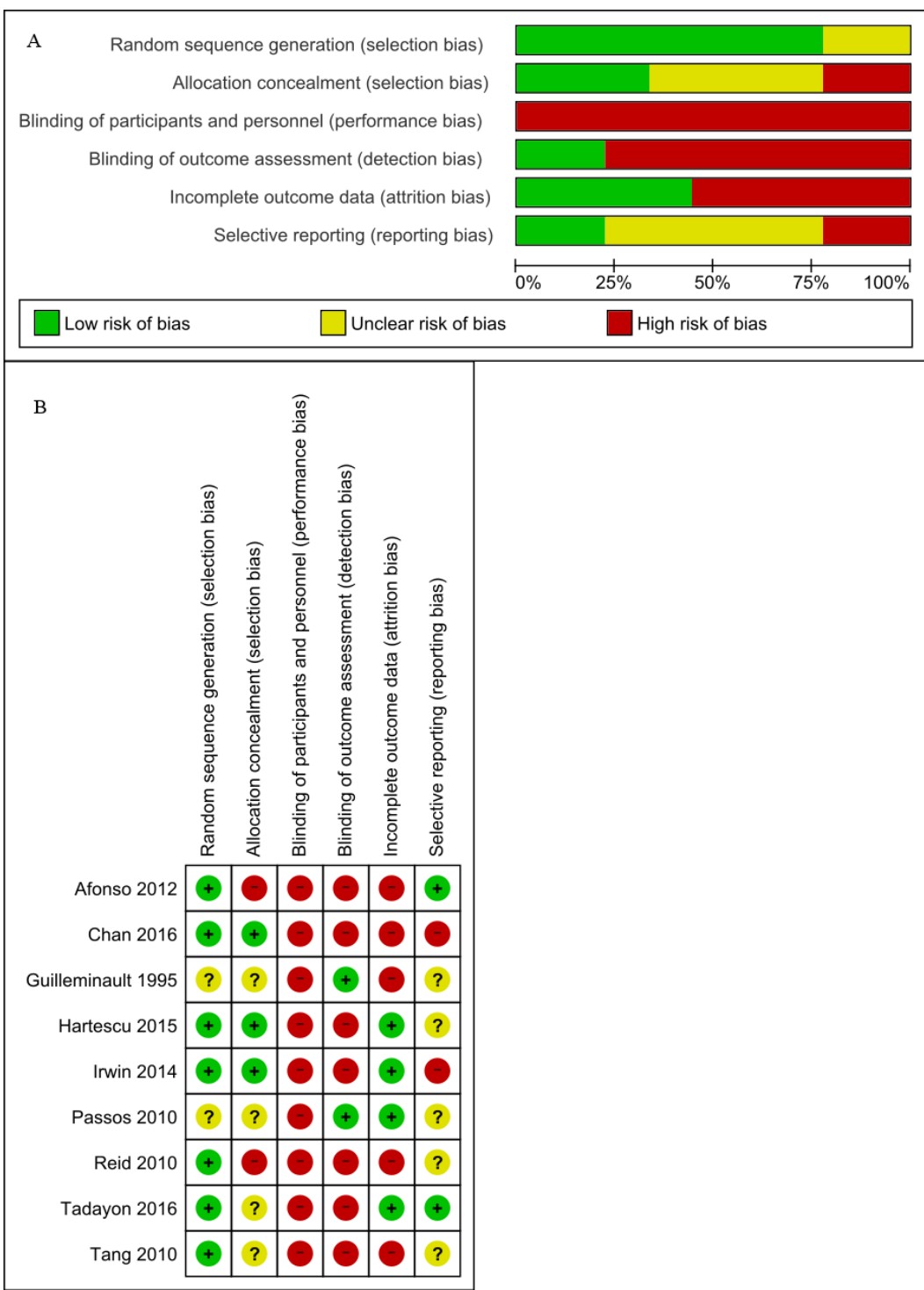

**Figure 2** **(A) Risk of bias graph (B) Risk of bias summary.** (A) Review author judgments about the risk for each bias item presented as percentages across all included trials. (B) Review author judgments about the risk for each bias item in all included trials.

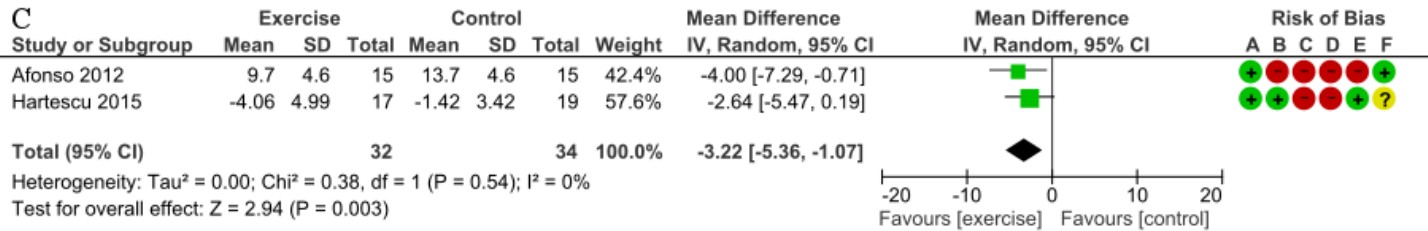

**A**

| Study or Subgroup | Exercise Mean | SD | Total | Control Mean | SD | Total | Weight | Mean Difference IV, Random, 95% CI |
|---|---|---|---|---|---|---|---|---|
| Chan 2016 | 8.6 | 3.5 | 27 | 9.1 | 3.3 | 25 | 14.7% | -0.50 [-2.35, 1.35] |
| Hartescu 2015 | -2.47 | 3.26 | 17 | 0.89 | 3.43 | 19 | 12.5% | -3.36 [-5.55, -1.17] |
| Irwin 2014 | 8 | 3.4 | 48 | 9.6 | 3 | 25 | 17.1% | -1.60 [-3.12, -0.08] |
| Reid 2010 | 5.1 | 1.44 | 10 | 9.5 | 1.97 | 7 | 15.7% | -4.40 [-6.11, -2.69] |
| Tadayon 2016 | 9.35 | 3.01 | 56 | 13.05 | 1.62 | 56 | 21.9% | -3.70 [-4.60, -2.80] |
| Tang 2010 | 9.78 | 3.06 | 36 | 13.11 | 2.89 | 35 | 18.1% | -3.33 [-4.71, -1.95] |
| **Total (95% CI)** | | | **194** | | | **167** | **100.0%** | **-2.87 [-3.95, -1.79]** |

Heterogeneity: Tau² = 1.18; Chi² = 15.45, df = 5 (P = 0.009); I² = 68%
Test for overall effect: Z = 5.21 (P < 0.00001)

Risk of bias legend
(A) Random sequence generation (selection bias)
(B) Allocation concealment (selection bias)
(C) Blinding of participants and personnel (performance bias)
(D) Blinding of outcome assessment (detection bias)
(E) Incomplete outcome data (attrition bias)
(F) Selective reporting (reporting bias)

**B**

| Study or Subgroup | Exercise Mean | SD | Total | Control Mean | SD | Total | Weight | Mean Difference IV, Random, 95% CI |
|---|---|---|---|---|---|---|---|---|
| Afonso 2012 | 78.1 | 9.1 | 15 | 82.3 | 8.6 | 15 | 20.4% | -4.20 [-10.54, 2.14] |
| Hartescu 2015 | 82.65 | 7.28 | 17 | 82.57 | 7.35 | 18 | 34.9% | 0.08 [-4.77, 4.93] |
| Irwin 2014 | 80.6 | 11.2 | 48 | 79.5 | 10.3 | 25 | 31.2% | 1.10 [-4.03, 6.23] |
| Passos 2010 | 81.4 | 11.8 | 36 | 81.9 | 12 | 12 | 13.5% | -0.50 [-8.31, 7.31] |
| **Total (95% CI)** | | | **116** | | | **70** | **100.0%** | **-0.56 [-3.42, 2.31]** |

Heterogeneity: Tau² = 0.00; Chi² = 1.74, df = 3 (P = 0.63); I² = 0%
Test for overall effect: Z = 0.38 (P = 0.70)

Risk of bias legend
(A) Random sequence generation (selection bias)
(B) Allocation concealment (selection bias)
(C) Blinding of participants and personnel (performance bias)
(D) Blinding of outcome assessment (detection bias)
(E) Incomplete outcome data (attrition bias)
(F) Selective reporting (reporting bias)

**C**

| Study or Subgroup | Exercise Mean | SD | Total | Control Mean | SD | Total | Weight | Mean Difference IV, Random, 95% CI |
|---|---|---|---|---|---|---|---|---|
| Afonso 2012 | 9.7 | 4.6 | 15 | 13.7 | 4.6 | 15 | 42.4% | -4.00 [-7.29, -0.71] |
| Hartescu 2015 | -4.06 | 4.99 | 17 | -1.42 | 3.42 | 19 | 57.6% | -2.64 [-5.47, 0.19] |
| **Total (95% CI)** | | | **32** | | | **34** | **100.0%** | **-3.22 [-5.36, -1.07]** |

Heterogeneity: Tau² = 0.00; Chi² = 0.38, df = 1 (P = 0.54); I² = 0%
Test for overall effect: Z = 2.94 (P = 0.003)

Risk of bias legend
(A) Random sequence generation (selection bias)
(B) Allocation concealment (selection bias)
(C) Blinding of participants and personnel (performance bias)
(D) Blinding of outcome assessment (detection bias)
(E) Incomplete outcome data (attrition bias)
(F) Selective reporting (reporting bias)

**Figure 3** **(A) Forest plot of comparison: Total PSQI score (B) Forest plot of comparison: Sleep efficiency (%) (C) Forest plot of comparison: Total ISI score.** (A) Total PSQI score was measured subjectively. IV, inverse variance; PSQI, Pittsburgh Sleep Quality Index (B) Sleep efficiency was measured objectively by the devices (e.g., PSG, actigraphy). IV, inverse variance; PSG, polysomnograph (C) Total ISI score was measured subjectively. ISI, Insomnia Severity Index; IV, inverse variance

## Secondary outcomes
### QOL
Five trials examined QOL, but the data were not subjected to the meta-analysis or assessed by the GRADE approach because concepts of QOL measures differed among the trials. The 12-item medical outcomes study short form health survey version 2.0 (SF-12v2) or SF-36 had two types of scores (physical component summary and mental component summary). Other QOL instruments had a single total score. Therefore, we did not calculate the SMD of the QOL instruments. Significant effects in favor of the intervention were noted in all trials (Figs. S1–S14; Table S4).

### Sleep onset latency
Data from five trials that measured sleep onset latency (min) in 206 participants were pooled for the meta-analysis (*Guilleminault et al., 1995*; *Passos et al., 2010*; *Afonso et al., 2012*; *Irwin et al., 2014*; *Hartescu, Morgan & Stevinson, 2015*). All trials measured sleep onset latency using PSG and actigraphy and had an intervention period of one day to six months. There was no significant improvement in favor of the intervention (MD, 1.90 min higher in the intervention group; 95% CI, 3.63 min lower to 7.43 min higher; $P = 0.50$; low-quality evidence). Shorter duration was more beneficial for sleep onset latency. No statistical heterogeneity was indicated (Tau$^2$ < 0.001; $I^2 = 0\%$) (Fig. S15; Table S4).

### Total sleep time
Data from five trials that examined total sleep time (min) in 206 participants were pooled for the meta-analysis (*Guilleminault et al., 1995*; *Passos et al., 2010*; *Afonso et al., 2012*; *Irwin et al., 2014*; *Hartescu, Morgan & Stevinson, 2015*). All trials measured total sleep time using PSG and actigraphy and had an intervention period of one day to six months. There was no significant improvement in favor of the intervention (MD, 4.32 min higher in the intervention group; 95% CI, 9.19 min lower to 17.84 min higher; $P = 0.53$; low-quality evidence). Longer duration was more beneficial for total sleep time. No statistical heterogeneity was indicated (Tau$^2$ < 0.001; $I^2 = 0\%$; Fig. S16; Table S4).

### All adverse events (defined by the trial list)
Four trials comprising 150 participants measured adverse events. Three trials found no adverse events in any of the participants (*Reid et al., 2010*; *Afonso et al., 2012*; *Chan et al., 2016*). One trial described one adverse event, a mild sprained ankle, in the intervention group (*Hartescu, Morgan & Stevinson, 2015*). Follow-up was two to six months (very low-quality evidence).

### Other secondary outcomes (Secondary outcomes not including Summary of findings table)
Anxiety and depression were significantly ameliorated in favor of the intervention (Figs. S17 and S18; Table S4). ESS and WASO did not detect a significant effect in favor of the intervention (Figs. S19 and S20; Table S4). None of the trials measured SSS (Fig. S21; Table S4).

## Additional analyses

We performed subgroup analyses of sleep quality because the outcome showed an $I^2 > 50\%$. We conducted an ad-hoc subgroup analysis for exercise frequency (acute or regular) because the underlying mechanisms may differ between acute exercise and regular exercise. We also conducted an ad-hoc subgroup analysis of background variables (cancer patients, postmenopausal women, and others). The exercise type subgroups differed significantly ($P < 0.001$; Table S4) and other pre-specified and ad-hoc subgroups of sleep quality did not differ significantly (Table S4). Sleep efficiency did not improve significantly in favor of the intervention with acute exercise (MD, 0.50% lower in the intervention group; 95% CI, 8.31% lower to 7.31% higher; $P = 0.90$) or regular exercise (MD, 0.56% lower in the intervention group; 95% CI, 3.64% lower to 2.52% higher; $P = 0.72$; Table S4). A higher percentage was more beneficial for sleep efficiency.

We conducted sensitivity analysis by restricting the analyzed studies to those that had a low risk of selection bias; however, the results were the same as those obtained in the original analysis (Table S4). Moreover, the results did not change with the use of a fixed-effects model instead of a random-effects model (Table S4). We were unable to estimate the ISI results, as none of the trials showed a low risk of selection bias (Table S4).

When studies using imputed data or per-protocol analysis were excluded, PSQI (two trials with 164 participants) did not exhibit a significant effect in favor of the intervention (MD, 2.21 points lower in the intervention group; 95% CI, 5.34 points lower to 0.92 point higher). A lower score was more beneficial for PSQI. Sleep efficiency (one trial with 48 participants) did not significantly improve in favor of the intervention (MD, 0.50% lower in the intervention group; 95% CI, 8.31% lower to 7.31% higher). A higher percentage was more beneficial for sleep efficiency. We were unable to estimate the ISI results because no trials remained after exclusion of those with imputed data or per-protocol analysis (Table S4).

We conducted an ad-hoc sensitivity analysis by excluding one study with acute exercise because it was an experimental RCT. When the study with acute exercise was excluded, sleep efficiency did not significantly improve in favor of the intervention (MD, 0.56% lower in the intervention group; 95% CI, 3.64% lower to 2.52% higher). A higher percentage was more beneficial for sleep efficiency.

## DISCUSSION

The pooled results revealed that exercise improves PSQI and ISI scores. These results were consistent across the included trials despite the indication of substantial heterogeneity in the PSQI. The heterogeneity of PSQI seemed to be explained by exercise type. Whether exercise improves QOL was inconclusive in our study, although exercise did have some adverse effects which were of little importance. These results suggested that exercise was an effective nonpharmacological treatment because improved sleep quality is one of the primary treatment goals (*Schutte-Rodin et al., 2008*). Furthermore, a recent comprehensive narrative review strongly recommended aerobic exercise in subjects with sleep disorders (*Chennaoui et al., 2015*). Exercise can be as promising a nonpharmacological intervention for patients with insomnia as CBT-I.

## Results compared to those of prior studies

A three-point change in PSQI score was chosen to indicate a minimal clinically important difference (MCID) (*Hughes et al., 2009*). Therefore, the effect of exercise on PSQI in favor of the intervention (low-quality evidence) was considered small. A previous study (*Yang et al., 2012*) found a small-to-moderate effect (SMD, 0.47; 95% CI [0.08–0.86]) of exercise on PSQI among patients with sleep complaints, whereas our study found that exercise exerts a large effect (SMD, 1.00; 95% CI [0.48–1.53]) on the PSQI. These results suggest that exercise may provide more beneficial effects on PSQI in patients with insomnia than in participants with sleep complaints. There is a possible ceiling and floor effect of exercise on sleep in patients with sleep complaints compared to those with insomnia (*Chennaoui et al., 2015*). For example, baseline total PSQI scores may be higher in patients with insomnia than in those with sleep complaints, which may explain the differences in the results of these studies.

Since a change in ISI score greater than 7 would be considered moderate improvement (*Morin et al., 2011*), the effect of exercise on ISI (MD, 3.22 points lower in the intervention group; 95% CI, 5.36 points lower to 1.07 points lower; very low-quality of evidence) in favor of the intervention was considered small. The only previous study using PSG (*Yang et al., 2012*) detected no change in sleep efficiency or onset latency, which was consistent with results on these two parameters in our study.

In the present study, exercise did not improve sleep efficiency, sleep onset latency, or total sleep time, and there was no evidence of heterogeneity across studies. The non-randomized crossover study demonstrated an acute morning exercise decrease in the arousal index and the number of stage shifts during the second half of the night in older individuals with insomnia (*Morita, Sasai-Sakuma & Inoue, 2017*). A polysomnographic and subjective sleep study found a significant decrease in sleep onset latency and wake time after sleep onset as well as a significant increase in sleep efficiency following a six-month exercise training program, but no significant differences were seen between morning and late-afternoon exercise in chronic primary insomnia (*Passos et al., 2011*). Inconsistent subjective and objective results regarding the effects of exercise on sleep, which may be related to variations in exercise intensity, and time between exercise and sleep, were reported. Moreover, acute exercise affects the endocrine system (*Tuckow et al., 2006*), metabolism (*Scheen et al., 1996*), and core body temperature (*Murphy & Campbell, 1997*; *Gilbert et al., 2004*). Regular exercise affects the endocrine system (*Kern et al., 1995*), metabolism (*Scheen et al., 1996*), circadian rhythm and body core temperature (*Murphy & Campbell, 1997*). Sleep loss may affect metabolism, the central nervous system, the endocrine system, inflammation, and the autonomic nervous system (*Chennaoui et al., 2015*). Some studies have focused on the sleep process in insomnia. Regular daytime exercise can increase melatonin secretion in and improve the sleep quality of patients with insomnia (*Taheri & Irandoust, 2018*). Insomnia can also result in cognitive dysfunction because sleep may restore cognitive function and maintain attentional mechanisms (*Taheri & Arabameri, 2012*). Thus, the beneficial effects of exercise on sleep efficiency and onset latency contribute to the interaction between

circadian rhythm and metabolic, immune, thermoregulatory, and endocrine effects. Future trials to investigate the effects of exercise on sleep cycle and sleep process in patients with insomnia are required.

### Summary of the findings and recommendatons

We first performed a systematic review and meta-analysis of the effects of exercise on sleep in patients with insomnia (diagnosed using criteria or screened with questionnaires). Our findings suggest that the effects of exercise on sleep were greater in patients with insomnia than in other populations and should be an effective nonpharmacological intervention. Exercise interventions may alleviate symptoms in patients with insomnia without use of hypnotics. The American Academy of Sleep Medicine report does not include exercise as a viable recommendation for treating insomnia (*Morgenthaler et al., 2006*). Our findings suggest that future clinical practice guidelines should include exercise as a recommendation for treating patients with insomnia.

### Strengths

The primary strength of this study was its careful and rigorous screening, extraction, and scoring process. The secondary strength was the extensive subgroup analyses that explored the heterogeneity of the results.

### Limitations

Our study has several limitations. First, only four of the nine included trials examined adverse effects (*Reid et al., 2010*; *Afonso et al., 2012*; *Hartescu, Morgan & Stevinson, 2015*). Therefore, unreported outcomes and important unmeasured outcomes such as adverse effects (for example, arrhythmia) may exist (*Andersen et al., 2013*). Second, most studies had a high or unclear risk of selection bias, although our sensitivity analysis revealed that the results were unchanged when studies were restricted to those that had a low risk of selection bias (Table S4). In the future, trials with low risks of selection bias need to be conducted verify our findings. Third, our review did not consider menopause in the meta-analysis because none of the included studies reported subgroup data by postmenopausal status. In the future, trials with subgroup data on postmenopausal women compared with women of other age groups are needed to determine the effects of exercise in patients with insomnia.

## CONCLUSIONS

Our findings suggest that exercise can improve sleep quality without notable adverse effects in patients with insomnia. Most of the trials included in our review suggested a high risk of selection bias in some domains. Therefore, higher quality research is needed to clarify the effects of exercise on sleep in patients with insomnia.

## ACKNOWLEDGEMENTS

We are grateful to Mr. Rui Afonso (Departamento de Psicobiologia, Universidade Federal de São Paulo, Sao Paulo), Prof. Helena Hachul (Departamento de Psicobiologia, Universidade Federal de São Paulo, Sao Paulo; Departamento de Ginecologia, Universidade Federal de

São Paulo, Sao Paulo), Dr. Iuliana Hartescu (Clinical Sleep Research Unit, Loughborough University, Loughborough), Dr. Arun Kumar (Department of Physiology, Seth G. S. Medical Collage & K. E. M. Hospital, Mumbai), Dr. Kathryn J. Reid (Department of Neurology and Center for Circadian and Sleep Medicine, Northwestern University Feinberg School of Medicine, Chicago, IL), Prof. Phyllis C. Zee (Department of Neurology and Center for Circadian and Sleep Medicine, Northwestern University Feinberg School of Medicine, Chicago, IL) and Assistant Professor Dr. Aileen WK Chan (The Nethersole School of Nursing, The Chinese University of Hong Kong, Hong Kong), the authors of the included study, for providing the detailed information. We thank Dr. Jessie Chan and Prof. Cecilia Chan from Department of Social Work and Social Administration, Centre on Behavioral Health, The University of Hong Kong, Hong Kong, for kindly providing valuable unpublished data. We are grateful to Dr. Kiyomi Shinohara (Department of Health Promotion and Human Behavior, Kyoto University Graduate School of Medicine/ School of Public Health, Kyoto) for helping to make scope of the review and providing information for PsycNet to enable screening in PsycINFO. We are grateful to Dr. Kazuhiro Uda (Office for Infectious Control, National Center for Child Health and Development, Tokyo) for helping collect references. We would like to thank Editage (http://www.editage.jp) for English language editing.

### Funding

This work was supported by Japan Society for the Promotion of Science (KAKENHI Grant Number 25282210), Nagoya University Academy of Psychiatry, and self-funding. The funders had no role in study design, data collection and analysis, decision to publish, or preparation of the manuscript.

### Grant Disclosures

The following grant information was disclosed by the authors:
Japan Society for the Promotion of Science: 25282210.
Nagoya University Academy of Psychiatry.

### Competing Interests

Masahiro Banno has received speaker honoraria from Dainippon Sumitomo, Eli Lilly, and Otsuka; honoraria for a manuscript from Seiwa Shoten Co., Ltd, SENTAN IGAKU-SHA Ltd and Kagakuhyoronsha Co., Ltd.; and travel fees from Yoshitomi Pharmaceutical Industries Ltd. Yuki Kataoka received research funds from Eli Lilly. The other authors declare no competing interests.

### Author Contributions

- Masahiro Banno, Hiraku Tsujimoto, Yasushi Tsujimoto and Yuki Kataoka conceived and designed the experiments, performed the experiments, analyzed the data, contributed reagents/materials/analysis tools, prepared figures and/or tables, authored or reviewed drafts of the paper, approved the final draft.

- Yudai Harada performed the experiments, authored or reviewed drafts of the paper, approved the final draft.
- Masashi Taniguchi, Ryo Tobita and Akiko Noda conceived and designed the experiments, authored or reviewed drafts of the paper, approved the final draft.

## Data Availability

The raw data are provided in a Supplemental File.

## Supplemental Information

Supplemental information for this article can be found online at http://dx.doi.org/10.7717/peerj.5172#supplemental-information.

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
