# Peer review of "Exercise can improve sleep quality: a systematic review and meta-analysis"

_PeerJ, doi:10.7717/peerj.5172_

## Round 0.1 · original submission · Major Revisions

The reviewers are generally supportive of your manuscript but have raised some questions that need to be addressed. I feel confident that you will be able to respond to each of their thoughtful suggestions and make edits to the manuscript as appropriate.

·

Basic reporting

None

Experimental design

Data collection and analysis
Usually two authors are required to do the screening the titles and abstracts. Why authors used 5 individual authors for screening?
Again in data extraction, only two authors are needed, why did you use 4?
Method: why authors used Random –effects model? While when we do not have heterogeneity more than 20%, using Fixed Effect is recommended.
Sensitivity analysis is when we have heterogeneity and in the forest plot or funnel plot we recognize one or two studies may cause the heterogeneity. In this case, we exclude one or two studies from analysis and run the meta- analysis.
The first line of meta-analysis is Fixed method and the second line is random effect, while authors did vice-versa.

Validity of the findings

Results: Sleep quality: the heterogeneity is too much; please take advantages from sensitivity analysis for excluding study/is which cause heterogeneity.
From Table 1: it seems that authors should have considered sub-group analysis for postmenopausal women versus women in other age, or people who had cancer and insomnia should consider as a subgroup.

Additional comments

None

·

Basic reporting

The introduction part seems to be too short far a systematic review. Its highly recommended to add more relevant references especially regarding the effects of different types of exercise in insomnia with emphasis on sleep cycle. the article literature doesn't have sufficient background to provide the broader field of knowledge. relevant literature must be referenced appropriately.
why haven you used PubMed as an important data base where you could find more relevant references. you have pointed out in the first line of abstract that "no systematic review have examined the effects of exercise on insomnia....", while in a very simple search, you can find something different as follows:
Effect of exercise on sleep quality and insomnia in middle-aged women: A systematic review and meta-analysis of randomized controlled trials.Maturitas. 2017 Jun;100:49-56. doi: 10.1016/j.maturitas.2017.04.003. Epub 2017 Apr 5.
its not possible to consider the insomnia while you haven't discussed the sleep process. in this regard, there are some more references helping your article to improve, so add them to your work as such:
1- The Effect of Sleep Deprivation on Choice Reaction Time and Anaerobic Power of College Student Athletes. Asian J Sports Med. 2012 Mar; 3(1): 15–20.
2- The Exercise-Induced Weight Loss Improves Self-Reported Quality of Sleep in Obese Elderly Women with Sleep Disorders. Sleep and Hypnosis: A Journal of Clinical Neurosciences and Psychopathology 2018;20(1):54-59
the rest parts including article structure, figs, tables and raw data shared seems to be appropriately prepared.

Experimental design

the original primary research is within Aims and Scope of the journal and research question has well been designed. all in all, this part is good

Validity of the findings

no more new technologies for insomnia monitoring has been considered in the research. the authors should focus more actigraphy results for the research question instead of PSQI scale. the following references can help well:
Utility of Actiwatch sleep monitor to assess waking movement behavior in older women. Med Sci Sports Exerc. 2014 Dec; 46(12): 2301–2307.
Measuring Sleep: Accuracy, Sensitivity, and Specificity of Wrist Actigraphy Compared to Polysomnography. Sleep. 2013 Nov 1; 36(11): 1747–1755.

Additional comments

Generally speaking, It is a good work, but needs some more relevant references focusing on exercise mechanisms as noticed above.
Good luck.

---

## Round 0.2 · Minor Revisions

Thank you for your extremely constructive responses to the reviewers' questions and requests. I'll leave it up to you whether or not you add more references as suggested by reviewer #2.

I have a small number of mostly very minor comments that I will suggest that you consider for the final version of your manuscript. These are almost all typographical comments or my thoughts around making the wording clearer where possible.

Line 41: "efficacy" refers to the effect of a treatment or intervention under ideal condition, whereas "effectiveness" refers to the same under real-world conditions. Interventions in this area will inevitably encounter compliance issues and so "effectiveness" is what is reported in manuscripts that do not provide a per protocol analysis. I had assumed that you used the effectiveness results (from an ITT analysis) where both ITT and PP analyses were presented and if this is indeed the case, "effectiveness" would be more appropriate. If you have a mixture, perhaps "effectiveness/efficacy" could be used instead?

Line 54: Rather than saying "did not improve", I think better would be "was not associated with improved" as a lack of evidence is not in itself proof of a lack of effect.

Line 55 and elsewhere: This is very much just a suggestion, but the negative signs in effects and CIs could be reduced if you regarded improvements as moving in the favourable direction. Sometimes this might require some clarification (e.g. "MD xx points higher in the intervention group; 95% CI xx to xx"), especially for non-statistically significant effects but it would make the results easier for readers I think. If you did do this, it would require careful checking of the results for negative signs.

Line 65: There is a spurious period after "Health" here.

Line 73: There is a missing space before "(CBT-I)".

Lines 100-102: I wonder if these references would flow better if placed immediately after the populations they refer to?

Line 154: I think "We excluded..." would be the start of another paragraph given the list items in the inclusion paragraph. Alternatively, you could combine the list items into the paragraph beginning on Line 146 so that all of Lines 146-155 were a single paragraph.

Lines 168, 170, and 177: I'd suggest deleting "was examined" so that all list items have similar structure.

Lines 213, 218, and 225. Reviewer #2 asked about this last time and your explanation is perfectly reasonable but I suspect that readers will ask the same question and suggest that you add a clarifying remark at both points, e.g. "with MB and one other author assigned to each article" (if I have understood the protocol correctly). Related to this, what happened if YK was one of the authors? Who then provided arbitration?

Line 232: It's not immediately clear when SMDs and MDs were used. The results text all present MDs and it is only in the Discussion that the PSQI result is presented as a SMD to compare with Yang, et al. Could this point be made clearer here?

Line 245: While anyone familiar with RevMan will appreciate that this involves DerSimonian and Laird's approach (at the moment, I believe, this is still the only option for a random effects meta-analysis using this software, but this could change in the future). This technical detail (the use of DSL’s approach) could be added here.

Lines 252-253: Please use a non-strict inequality here to indicate which group exactly 6 months would fit into.

Lines 259-260: It's not the low risk of adequate random sequence generation you're worried about, perhaps: "low risk for both random sequence generation and random allocation"?

Line 262: I'd suggest deleting "with a per-protocol or intention-to-treat analysis" here as this is likely to be a given in this context that studies will be one or the other.

Line 263: Note that this should be "imputed data" rather than "imputed statistics". Same point on Lines 356 and 360.

Lines 293, 300, 307, 320, 328, 349, 350, and potentially elsewhere: I'd be happy not to have the z-value presented here. It's more a means of calculating the p-value and given the p-value (to sufficient precision) and the direction of the effect, it could be calculated anyway but I'd be surprised if readers actually paid it any attention.

Line 313: "concepts" isn't very clear to me here. Can you explain this in another way? SMDs would have been an option for analysing different QoL instruments as long as the construct was the same or at least sufficiently similar.

Line 320: It might be a trailing zero but "1.9 min" should have 2 decimal places to be consistent with other results. If you preferred to show this with one decimal place, that would be fine also as long as all results in minutes were the same.

Line 390: "on the MCID" seems out of place here and could be deleted.

Line 395: Perhaps "In the present meta-analysis"? This is entirely up to you.

Line 396: I'd be inclined to say "no evidence of heterogeneity" as while the I-squared values were zero, their CIs would extend to non-zero values.

Line 420: The paragraph listing strengths is more a summary of the findings and recommendations rather than strengths (which are reasons why the reader should feel confident in your results, the inverse of the limitations which are reasons why the reader might feel cautious about your findings). Note that novelty is not a strength but there are definitely strengths to the work you have done. Perhaps discuss the careful and rigorous screening, extraction, and scoring processes instead here? The extensive subgroup analyses also allowed you to explore some of the heterogeneity in the results. These are merely suggestions and you might like to look at other articles to get some more ideas of what might be worth mentioning here.

Figure 1: Even as an eps file, this still looked "blocky" to me. Assuming this isn't only an issue for me, could you provide a higher quality version of this file? I'd suggest deleting the ", with reasons" from the three boxes on the right as I interpret these as instructions rather than content to be retained in the diagram.

Table 1 summary. Note that "alfa" should be "alpha". This is also the case on Line 19 of Page 45.

·

Basic reporting

No comments

Experimental design

No comments

Validity of the findings

No comments

Additional comments

Nothing

·

Basic reporting

ok

Experimental design

ok

Validity of the findings

ok

Additional comments

Dear Authors
Thanks for replying the previous comments
one major point is not well considered in your research. ac cited before, your title refers to considering the sleep quality, and neither specific age group nor population hasn't been categorized to be considered, so, its highly recommended to consider two main categories (elderly people and athletes) who have been enormously studied in different investigations. since exercise plays the key role in sleep cycle of athletes and elderly, more relevant references in this context must be applied, therefore the following reference are also recommenced to be referred.
1- Monleon C, Hemmati Afif A, Mahdavi S, Rezaei M. The Acute Effect of Low Intensity Aerobic Exercise on Psychomotor Performance of Athletes with Nocturnal Sleep Deprivation. Int J Sport Stud Hlth. 2018;1(1):e66783.
2- Irandoust K, Taheri M. The Effect of Strength Training on Quality of Sleep and Psychomotor Performance in Elderly Males. Sleep and Hypnosis. 2018;20(3):160-5.
3- Taheri M, Valayi F. Aerobic Exercise Improves Attention and Quality of Sleep Among Professional Volleyball Players.

---

## Round 0.3 · Minor Revisions

Thank you very much for your constructive revisions.

I apologise for my pedantry, but I still have one very minor comment. In your response, you note my comment about the low risks below.

"Lines 259-260: It's not the low risk of adequate random sequence generation you're worried about, perhaps: "low risk for both random sequence generation and random allocation"?"

And responded:

"Response: We were worried about both adequate random sequence generation and adequate random allocation.
We revised the expressions to clarify our intention as follows:
“low risks of both adequate random sequence generation and adequate random allocation”"

My point wasn't as clear as it should have been. The current text:

"repeating the analysis but restricting it to studies with low risks of both adequate random sequence generation and adequate random allocation"

could potentially be misread as including studies with low risk of adequate random sequence generation and allocation concealment (both of which are good things and so we'd prefer a high risk of these). What we don't want here are high risks of *bias* from these things. Perhaps:

"repeating the analysis but restricting it to studies with low risks of bias from random sequence generation and allocation concealment"

would make this point absolutely clear to the reader and better match your text later on lines 377-378:

"We conducted sensitivity analysis by restricting the analyzed studies to those that had a low risk of selection bias"

I'm very happy with all of the other revisions you have made and will be delighted to accept the manuscript once you've had a chance to consider revising the wording on this very minor point.

---

## Round 0.4 · accepted · Accept

Thank you for revising that one sentence, and also for your constructive revisions and responses along the way. Well done.